# Predicting the Impact of Diffuse Alveolar Damage through Open Lung Biopsy in Acute Respiratory Distress Syndrome—The PREDATOR Study

**DOI:** 10.3390/jcm8060829

**Published:** 2019-06-11

**Authors:** Pablo Cardinal-Fernandez, Guillermo Ortiz, Chih-Hao Chang, Kuo-Chin Kao, Emmanuelle Bertreau, Carole Philipponnet, Víctor Manuel Casero-Alonso, Bertrand Souweine, Emmanuel Charbonney, Claude Guérin

**Affiliations:** 1Hospital Universitario HM Sanchinarro, 28050 Madrid, Spain; pablocardinal@hotmail.com; 2Fundación HM, 28050 Madrid, Spain; 3Universidad del Bosque, Bogotá #131a2, Colombia; ortiz_guillermo@hotmail.com; 4Universidad de Barcelona, 08007 Barcelona, Spain; 5Chang Gung Memorial Hospital, Taoyuan 33005, Taiwan; ma7384@cgmh.org.tw (C.-H.C.); kck0502@cgmh.org.tw (K.-C.K.); 6Réanimation Médicale Hôpital de la Croix Rousse, 69004 Lyon, France; emmanuelle.bertreau@chu-lyon.fr; 7Réanimation Médicale, F63000 Clermont-Ferrand, France; cphilipponnet@chu-clermontferrand.fr (C.P.); bsouweine@chu-clermontferrand.fr (B.S.); 8Statistic Area, Universidad de Castilla-La Mancha, 13071 Ciudad Real, Spain; VictorManuel.Casero@uclm.es; 9Centre de Recherche Hôpital du Sacré-Cœur de Montréal (HSCM), 5400 boul. Gouin Ouest, Montréal, QC H4J 1C5, Canada; emmanuel.charbonney@umontreal.ca; 10Lyon East Faculty of Medicine, University of Lyon, 69100 Lyon, France; 11INSERM, 955 Créteil, France; 12Hospices Civils de Lyon, 69003 Lyon, France

**Keywords:** acute respiratory distress syndrome, diffuse alveolar damage, acute respiratory failure

## Abstract

The aim of this retrospective and international study is to identify those clinical variables associated with diffuse alveolar damage (DAD), and to explore the impact of DAD on hospital mortality risk. Inclusion criteria were: adult patients with acute respiratory distress syndrome (ARDS) undergoing open lung biopsy (OLB) during their intensive care unit (ICU) management. The main end-points were: DAD and hospital mortality. In the training (*n* = 193) and validation cohorts (*n* = 65), the respiratory rate (odd ratio (OR) 0.956; confidence interval (CI) 95% 0.918; 0.995) and coronary ischemia (OR 5.974; CI95% 1.668; 21.399) on the day of ARDS had an average area under the receiver operating characteristic curve (AUROC) of 0.660 (CI95% 0.585; 0.736) and 0.562 (0.417; 0.706), respectively. PEEP (OR 1.131; CI95% 1.051; 1.218) and coronary ischemia (OR 6.820; CI95% 1.856; 25.061) on the day of OLB had an average AUROC of 0.696 (CI95% 0.621; 0.769) and 0.534 (CI95% 0.391; 0.678), respectively, to predict DAD. DAD (OR 2.296; CI95% 1.228; 4.294), diabetes mellitus requiring insulin (OR 0.081; CI95% 0.009; 0.710) and the respiratory rate (OR 1.045; CI95% 1.001; 1.091) on the day of ARDS had an average AUROC of 0.659 (CI95% 0.583; 0.737) and 0.513 (CI95% 0.361; 0.664) to predict hospital mortality and DAD (OR 2.081; CI95% 1.053; 4.114), diabetes mellitus requiring insulin (OR 0.093; CI95% 0.009; 0.956), PaCO_2_ (OR 1.051; CI95% 1.019; 1.084), and platelets count (OR 0.999; CI95% 0.999; 0.999) the day of OLB had an average AUROC of 0.778 (CI95% 0.710; 0.843) and 0.634 (CI95%0.481; 0.787) to predict hospital mortalty in the training and validation cohorts, respectively. In conclusion, DAD could not to be predicted clinically and was significantly associated with hospital mortality.

## 1. Introduction

Acute respiratory distress syndrome (ARDS) remains a challenge for intensivists given its incidence (almost 10% of all intensive care unit (ICU) admissions) and high mortality rate [1]. The Berlin ARDS definition has acknowledged diffuse alveolar damage (DAD) as the pathological hallmark in the acute phase [2]. However, several studies on patients [3,4,5] or autopsies [6] have reported that only half of the subjects with ARDS actually had DAD. Thus, exploring the correlation between clinical and pathological diagnosis is important because: (a) ARDS outcome has been shown to be dependent on the presence or absence of DAD [3,4,5,7]; (b) the prevalence of DAD may explain why the beneficial effects of most pharmacological treatments in animal models [8] have not been observed in clinical practice [9,10,11]; and (c) several non-DAD pathological patterns correspond to diseases with specific treatments [7,10,12].

Tissue sampling is required to make the diagnosis of DAD. This can only be done with open lung biopsy (OLB). However, given the risks of the procedure [13,14], its indications are very restrictive [15,16] and only skilled practitioners and proper facilities can provide a reasonable benefit-to-risk ratio [4,5,11,17,18,19]. Biomarkers are an alternative to OLB. For example, procollagen III performed well to predict the fibro-proliferative stage of OLB in a cohort of 32 patients with persistent ARDS [20]. Currently, only one predictive model for DAD has been reported [6], but it is limited as it is based on autopsy data. On the other hand, the effect of DAD on mortality has been suggested in several studies [5,17,21,22] and in a meta-analysis [3]. However, the confounding variables were taken into account in only one of these studies [4]. This study proposed a predictive model of mortality, unfortunately, the discriminatory capability of this model was not reported and it was not validated in an independent cohort. Finally, steroid-sensitive histological patterns of OLB could have a better outcome than steroid-resistant histological patterns, including DAD [7]. All of the above data were obtained from small series of patients, and hence, there is a need to develop a model to predict DAD, as well as to define its impact on ARDS outcome.

With this rationale in mind, we undertook the present study. Our primary hypothesis was to explore whether DAD could be better predicted from data recorded at the time of ARDS or at the time of OLB. We based this reasoning on the fact that ARDS with DAD constitutes a clinical-pathological entity different from ARDS without DAD [2,4,5,6]. Our secondary hypothesis was that DAD is associated with an increased risk of ARDS mortality. This hypothesis is in line with the results of previous studies [4,5]. However, the external validation of this result is low because the studies were single-center or not based on patient data [3,4,5].

## 2. Methods

The protocol was approved by an ethics committee (Comité Consultatif de Protection des personnes Sud-Est II) of the Lyon University Hospital, Lyon, France on 10 October 2016 (IRB number 00009118). Informed consent from the patients or their next of kin was waived.

First of all, centers that performed OLB in patients with ARDS were approached. Among them, five agreed to participate. Then, an online meeting was organized with the aim to define: (a) inclusion and exclusion criteria; (b) variables to be retrieved; (c) time-point for variables; and (d) pathological definitions. Part of the data used for the purpose of the present study has been previously reported [4,5,17,18].

Patients were included if they met all the following criteria: (a) 18 years or more in age; (b) diagnosis of ARDS according to the Berlin definition [2]; and (c) OLB was performed during their ICU stay. Patients were not included if they did not meet ARDS criteria according to the Berlin definition or if lung histology was obtained by other means. A specific case record form was set out and included a series of variables recorded at the time of ARDS diagnosis and of the OLB (Appendix A) for each enrolled patient. The DAD definition used in the present study is that used in the corresponding previous reports: the presence of edema, hyaline membranes lining alveoli, and interstitial acute inflammation [4,5]. Other histological patterns were considered as non-DAD.

## 3. Data Analysis

Values were expressed as median (first-third quartiles) and count (percentage), unless otherwise stated. Comparisons between groups were performed using nonparametric tests (Mann-Whitney test or Chi-squared Test) as required.

The primary end-point was DAD on OLB. The secondary end-point was hospital mortality.

The specific statistical analysis was carried out using the following steps (Appendix A).

First, to validate each model, the whole database was randomly divided into two independent cohorts: a training cohort that accounted for 75% of all the patients and a validation cohort that included the remaining 25%.

Second, to maintain the power of the study and to reduce the selection bias sourced from excluding patients with missing data (NA), we conducted a multivariate imputation by chained equations (MICE) procedure [23,24]. MICE is a well-accepted procedure that estimates the most probable value for missing data based on the observed data. Likewise, this procedure allows us to set a starting point (“seed”), in this study the number 2690, to ensure that every time the imputation algorithm is running it produces the same database. As any missing value cannot be known with a 100% accuracy, MICE generates several datasets (in this manuscript five datasets per each missing value) that represent the uncertainty of the original value. Each data set differed only for NAs values and was analyzed independently for each [25]. MICE includes three steps: multiple imputations, analysis and pooling.

Third, a pooled logistic regression model (PLRM) was used separately for DAD and hospital mortality prediction. For that, the variables included in the model were automatically pooled over all the data sets. This procedure resulted in a unique PLRM that considers all significant data from all datasets [26]. As we created five data sets, we included in the so-called maximum PLRM all those variables, which, in at least one of five data sets exhibited a *p* value <0.1. To help the reader to better handle the manuscript, only the results from the first data base were included in the main text. The other results (databases 2 to 5) are shown in the online material. This strategy explains why some not statistically significant variables were included in the PLRM. Then, the variables with the highest *p* value were removed one by one from the model until all the variables in the model had a *p* value <0.05 (backward step procedure) (Appendix A). The odds ratio (OR) and 95% confidence intervals (CI) were used to quantify the strength of the association between covariates and dependent variable.

Fourth, we calculated the area under receiver operating characteristic curve (AUROC), the accuracy, sensitivity, and specificity, and the positive and negative likelihood for the PLRM in each dataset. The performance of the five datasets for each result was calculated. Finally, we also tested our validation cohort with both Lorente et al.’s autopsy model [6] to predict DAD and Kao et al.’s model [4] to predict hospital mortality.

The statistical analysis was performed using R software (R: a language and environment for statistical computing. R Foundation for Statistical Computing, Vienna, Austria, URL http://www.R-project.org) and the CBCgrps [27] and MICE [26] libraries.

## 4. Results

The 258 patients with ARDS (31 mild, 125 moderate, 102 severe) were randomly divided into a training (*n* = 193) and a validation (*n* = 65) cohort. OLB was performed 6 (2–12) days after admission into the ICU. No NA was found in the categorical variables, but several continuous variables were incomplete. The proportion of NAs between the training and validation cohort was similar (Appendix A). After imputation, both cohorts were similar (see Table 1 and Table 2 for dataset 1 and Appendix A for the datasets 2 to 5). Results pertaining to dataset 1 are shown in the main text, datasets 2–5 are available in the Appendix A.

DAD was observed in 143 of 258 patients (55%) with no difference between the training and the validation cohorts (56% vs. 52%, respectively, *p* = 0.68). Hospital mortality was observed in 158 of 258 patients (61%) with no difference between the training and the validation cohorts (61% vs. 63%, *p* = 0.84).

### 4.1. Diffuse Alveolar Damage Prediction

In the training cohort, coronary ischemic disease, home oxygen supplementation, plateau pressure, positive end-expiratory pressure (PEEP), PaO_2_/FiO_2_ ratio, respiratory rate, temperature, Index Normalized Ratio and dobutamine dose were significantly different between patients with (*n* = 84) and without DAD (*n* = 109) at the time of ARDS diagnosis (Table 3 and Table 4, and Appendix A). At the time of OLB, the corresponding variables were coronary ischemic disease, home oxygen supplementation, chronic use of inhaled steroids, plateau pressure, PEEP and number of affected quadrants in the chest X-ray (Table 3 and Table 4, and Appendix A).

The PLRM using the data from the day of the ARDS diagnosis identified respiratory rate OR 0.95; CI95% 0.918–0.995; *p* = 0.029) and coronary ischemia (OR 5.974; CI95% 1.668–21.339; *p* < 0.001) as independent covariates associated with DAD (Table 5). The PLRM using data from the day of the OLB identified PEEP (OR 1.131; CI95% 1.051–1.218; *p* < 0.001) and coronary ischemia (OR 6.820; CI95% 1.856–25.061; *p* < 0.001) as independent covariates associated with DAD (Table 5). AUROC, accuracy, sensitivity, specificity and likelihood ratios from the DAD predictive models and from the autopsy model developed by Lorente et al. [6] are shown in Table 6 (Figure 1).

### 4.2. Mortality Prediction

In the training cohort, the presence of DAD at the OLB and diabetes mellitus requiring insulin, age, static compliance, plateau pressure, PEEP, PaO_2_/FiO_2_ rate, driving pressure, respiratory rate, temperature, number of quadrants involved in the chest X-ray, heart rate and platelet counts at the time of ARDS diagnosis, were different between survivors (*n* = 76) and non survivors (*n* = 117) at the time of hospital discharge (Table 3 and Table 7, and Appendix A). The corresponding variables were DAD on OLB and diabetes mellitus requiring insulin, age, tidal volume, static compliance, driving pressure, plateau pressure, PaCO_2_, arterial pH, hemoglobin, leukocytes, platelets, total bilirubin, creatinine, body temperature and antiviral drugs administration at the time of OLB (Table 3 and Table 6, and Appendix A).

The PLRM using data from the day of ARDS diagnosis found that DAD (OR 2.296 [1.228–4.294]; *p* < 0.001), diabetes mellitus requiring insulin (OR 0.081 [0.009–0.710]; *p* = 0.023) and respiratory rate (OR 1.045 [1.001–1.091; *p* = 0.046] were independently associated with mortality (Table 5). The PLRM using data from the day of OLB, found that presence of DAD (OR 2.081 [1.053–4.114]; *p* = 0.035), diabetes mellitus requiring insulin (OR 0.093 [0.009–0.956]; *p* = 0.046), PaCO_2_ (OR 1.051 [1.019–1.084]; *p* < 0.01) and platelets count (OR 0.999 [0.999–0.999]; *p* < 0.001) were independently associated with mortality. AUROC, accuracy, sensitivity, specificity and likelihood ratios from mortality predictive models and from the mortality model developed by Kao Kum et al. are showed in Table 6 (Figure 1).

## 5. Discussion

The main findings of the present study were that DAD could not be predicted clinically (only the AUROCs using the trainee cohort are better than chance) and was significantly associated with hospital mortality (DAD was associated with an increased risk of death in PLRM at the time of both the ARDS and the OLB days).

Before discussing the present results, a critique of our methodology is required. We designed a multicenter, international, retrospective, observational study of a niche population. These qualifiers obviously explain some of the limitations of our study, in particular the issues related to missing data, whose statistical management is discussed below, and the generalizability of the findings.

According to Depuydt et al. [28], defining a disease requires clinical findings that should be linked, and associated to a specific histological pattern. In other words, the syndrome and the histological patterns could be found in different scenarios, but, when both are present simultaneously this defines a disease. The statistically significant association between DAD and hospital mortality after adjustment for confounding factors is in line with other reports [3,4], and constitutes a strong reason to consider the association of ARDS to DAD as a unique clinical-pathological entity that could be named “real” ARDS. The clinical picture of ARDS without DAD should be considered as an imitator [11,12,29]. However, the fact that DAD could not be predicted clinically rises a question: is it possible to define a disease when the histology cannot be easily diagnosed? From our point of view, the answer would be affirmative as this is what happens with almost all diseases (e.g., electrocardiogram findings or troponins have been reported dozens of years after the first description of the myocardial infarct), and reflects the lack of availability of a diagnostic tool. Furthermore, including the pathological pattern in the definition of ARDS would increase the accuracy of the outcome prediction, and modify the management of patients with ARDS [5,18,22,30]. For example, Gerard et al. [7] demonstrated that ARDS patients with a pathological pattern resistant to steroids (e.g., DAD) had a worst outcome as compared to those with a steroid-sensitive pattern (e.g., cryptogenic organizing pneumonia and bronchiolitis obliterans with organizing pneumonia, acute eosinophilic pneumonia and alveolar hemorrhage). The key point is that most of the sensitive-steroid patterns cannot be clinically distinguished from steroid-resistant patterns, which prevents clinicians from using specific treatments. Unfortunately, we could not replicate Gerard et al.’s [7] approach because we only summarized the OLB histology as DAD and non-DAD. Finally, several physio-pathological mechanisms and treatments have been successfully tried in animal models of ARDS but their translation to humans has failed. This may be explained by the fact that derangement in molecular pathways determines the pathology, and the pathology is a requirement in most of the ARDS animal models but not in human patients [11,31,32]. Indeed, in ARDS patients, entities like DAD, pneumonia with DAD, pneumonia, pulmonary fibrosis, normal lungs, as examples, [3,4,5,6], cannot be clearly identified as such without histological results.

In line with Park et al. [33] and previous studies [4,5,17,18], our present findings confirm that DAD could not be predicted using clinical variables. It is well known that ARDS susceptibility and prognosis are associated with well-known risks and protective factors [34,35,36]. However, we have identified factors associated with either a harmful (coronary ischemia and PEEP) or a protective (respiratory rate) effect regarding DAD in living patients with ARDS. Our results are different from those of Thille et al. [37], who reported that a duration of evolution of ARDS of more than 3 days, severe hypoxemia, increased dynamic driving pressure, and diffuse opacities involving the four quadrants on chest radiographs are associated to DAD in autopsies. Based on autopsy data, the prediction model for DAD by Lorente et al. [6] was similar to chance, as in our study. Taken together, these discrepant results may stem from the differences in design, i.e., autopsy studies vs. OLB performed during patient management. The significant association between coronary ischemia and DAD has never been reported before. It could represent either an underlying mechanism, as yet unknown, which should be further investigated, or a confounding factor (the real risk factor is the one associated to both the presence of coronary ischemia and DAD). We cannot exclude the random effect.

We found that diabetes reduced and hypercapnia increased the risk of hospital mortality. Diabetes has been reported as a protective factor for ARDS [38]. Regarding hypercapnia, our results are in line with a recent report that reinforced the association of increased PaCO_2_ and ARDS mortality [39].

Unexpectedly, the driving pressure, the stronger predictor of ARDS mortality in recent studies [40,41] and shown as a risk factor for DAD in autopsied patients [37], was neither associated with DAD nor with mortality in our cohort. We speculate that this result may be related to the selection bias of the present study, which included the most severe ARDS patients with a baseline high driving pressure (Appendix A). However, we found that the respiratory rate was associated with mortality but not with DAD. This result is important because it may have a connection to mechanical power. This is a new concept for explaining ventilator-induced lung injury [42,43]. It quantifies the relative weight of different components that may affect the lung during mechanical ventilation. Our results suggest that the expected increase in mechanical power from higher respiratory rate was not associated with DAD, but with mortality.

The impact of DAD on ARDS outcome and the fact that DAD cannot be predicted, either using an autopsy model [6] or with our data, highlight the need for further research based on other means that can be performed at large scale. There are different ways including biomarkers, lung imaging or genetic traits [34,35]. Based on our results, a study based on patients with ARDS and with an available histology is required. To the best of our knowledge, the only biomarker developed using lung histology from patients with ARDS is the alveolar procollagen III [20]. Novel, minimally invasive methods that use distal sampling of the airspace could be an avenue [44]. Pathology findings coupled with this could lead to personalized ARDS treatments [31,35]. The latter reinforces the pressing needed to identify a specific biomarker for DAD [29,31,44]. Finally, it is worth highlighting that several recent studies have reported that ARDS includes two different endophenotypes distinguished by their inflammatory profile (the hyper-inflammatory is characterized by high plasma levels of interleukin (IL)-6, IL-8, soluble tumor necrosis factor (TNF) receptor 1 and low protein C, and high prevalence of shock and metabolic acidosis, and the hypo-inflammatory subphenotype is characterized by lower levels of the inflammatory biomarkers, less acidosis and less vasopressor-dependent shock) and treatment response (PEEP, fluid management and simvastatin therapy [45]), which may reflect the effect of a variable that cannot be directly identified (a latent variable) [46,47]. Although the latent variable could be any feature or process (e.g., a physio-pathological pathway, a genetic trait, a molecular pattern, etc.), based on our results the main candidate for a latent significant variable should be DAD. We could not address the relation between endophenotypes and DAD as the former are mainly based on serum biomarkers (vide supra), which were not measured in our study.

Our study has several limitations. First, as for any retrospective study, the presence of missing values may have biased or reduced the power of the results. In this study, we have applied a validated strategy to reduce the impact of this limitation. As the proportion of a missing variable that can be imputated is unknown, we a priori decided to imputate all variables. Therefore, the effect of imputated data on the final results should be small as the proportion of NA variables independently associated to DAD or mortality are less than 12%. Second, there is an obvious selection bias since OLBs were only performed in non-resolving and severe ARDS at different time points. Selection bias is a frequent limitation in non-randomized studies and is associated with reduced representation of the sample as well as uncommon results, such as the high mortality rate found in our study. Third, the criteria for performing OLB may differ between centers (the specific reason for performing the OLB in each patient was not available). This heterogeneity may explain the time lag between ARDS diagnosis and OLB. Fourth, as DAD was recorded as a dichotomous variable, the effect of other histologic patterns was lacking. Likewise, we could not investigate the DAD phases (acute, fibroproliferative and fibrosis). It should be mentioned that according to the definition of DAD we used, the early acute phase of DAD was likely sampled. Fifth, 25% of the OLBs were performed within the first two days after ARDS diagnosis. This short time period may be not enough for DAD to develop and predictive variables may differ in a short-term OLB compared with an OLB done later. Sixth, we have analyzed the total respiratory rate (sum of spontaneous and assisted breath). Likewise, its discrepant effect on DAD and mortality may be either true or a false positive result (e.g., low power of the study, presence of a confounding variable or random effect). Finally, technical details for OLB were not recorded and may vary between centers. However, our study have several strengths. First, it analyzed at the patient’s level, the largest cohort of ARDS who underwent an OLB during their ICU stay. Second, the multicenter origin of the data from three different continents strengthen the validity of our analysis. Third, we used the same ARDS and DAD definition for all the patients. Fourth, we have compared our model with previously reported models. Our two predictive models (one at the time of ARDS diagnosis and the other at the time of OLB) showed an acceptable discriminatory ability and are better than the previous model described by Kao et al. [4].

## 6. Conclusions

In conclusion, this study confirms that histological DAD cannot be predicted based on clinical parameters, and that DAD is associated with a two-times higher risk of hospital mortality. Given the complexity and risks of OLB as a routine procedure, it is essential to develop alternate methods for diagnosing DAD, such as biomarkers or lung imaging techniques.

## Figures and Tables

**Figure 1 jcm-08-00829-f001:**
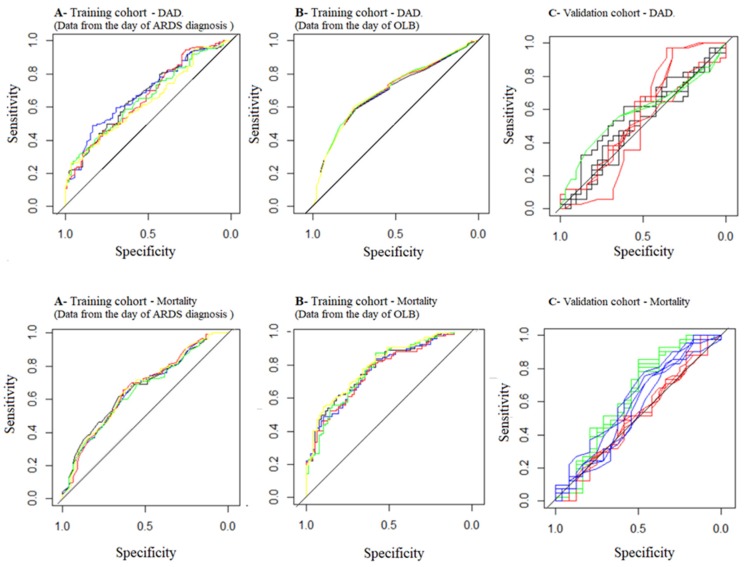
Receiver operating curves (ROC) for predicting diffuse alveolar damage. Upper line diffuse alveolar damage. (**A**) Training cohort. Data from the day of ARDS diagnosis (black: dataset 1; blue: dataset 2; red: dataset 3, green: dataset 4 and yellow: dataset 5). (**B**) Training cohort. Data from the day of OLB (black: dataset 1; blue: dataset 2; red: dataset 3, green: dataset 4 and yellow: dataset 5). (**C**) Validation cohort. The five data sets from each model are painted with the same color (black: Lorente’s model, red: day of ARDS, green: day of OLB). Lower line mortality. (**A**) Training cohort. Data from the day of ARDS diagnosis (black: dataset 1; blue: dataset 2; red: dataset 3, green: dataset 4 and yellow: dataset 5). (**B**) Training cohort. Data from the day of OLB (black: dataset 1; blue: dataset 2; red: dataset 3, green: dataset 4 and yellow: dataset 5). (**C**) Validation cohort. The five dataset from each model are painted with the same color (red: day of ARDS, green: day of OLB, blue; Kao et al. model).

**Table 1 jcm-08-00829-t001:** Comorbidities and baseline characteristics of 258 patients with ARDS in the training and validation cohorts.

Comorbidities	All Patients (*n* = 258)	Training (*n* = 193)	Validation (*n* = 65)	*p* Value between Training and Validation
Active smoking ^†^	42 (16)	37 (19)	5 (8)	0.048
Active solid neoplasms ^†^	48 (19)	37 (19)	11 (17)	0.827
Active hemato-oncology neoplasm ^†^	39 (15)	32 (17)	7 (11)	0.352
Chemotherapy within the last 3 months ^†^	29 (11)	22 (11)	7 (11)	1.000
Organ transplant ^†^	8 (3)	7 (4)	1 (2)	0.684
Acute immunodeficiency syndrome/human immunodeficiency virus ^†^	10 (4)	7 (4)	3 (5)	0.716
Arterial hypertension ^†^	71 (28)	55 (28)	16 (25)	0.656
Coronary ischemia ^†^	28 (11)	22 (11)	6 (9)	0.798
Chronic cardiac failure (NYHA III or IV) ^†^	29 (11)	22 (11)	7 (11)	1.000
Chronic kidney injury ^†^	13 (5)	11 (6)	2 (3)	0.527
Diabetes mellitus ^†^	46 (18)	35 (18)	11 (17)	0.973
Diabetes mellitus requiring insulin ^†^	8 (3)	8 (4)	0 (0)	0.208
Chronic obstructive pulmonary disease ^†^	48 (19)	35 (18)	13 (20)	0.881
Domiciliary oxygen therapy ^†^	10 (4)	8 (4)	2 (3)	1.000
Inhaled β2 agonist ^†^	6 (2)	6 (3)	0 (0)	0.342
Inhaled steroids ^†^	11 (4)	11 (6)	0 (0)	0.071
Systemic steroids ^†^	27 (10)	26 (13)	1 (2)	0.013
**Baseline Characteristic**				
Female gender ^†^	89 (34)	66 (34)	23 (35)	0.981
Age (years) ^‡^	62 (46; 71)	61 (46; 70)	64 (48; 74)	0.372
Weight (Kg) ^‡^	65 (57; 76)	65 (57; 75)	69 (60; 77)	0.343
Diffuse Alveolar Damage ^†^	109 (55.4)	109 (56.6)	34 (52.3)	0.656
Days from ARDS to OLB ^‡^	6 (2; 12)	6 (2; 12)	6 (2; 14)	0.819
Days from hospital admission to ARDS ^‡^	3 (1; 9)	3 (0; 90)	4 (1; 80)	0.790
Length of hospital stay (days) ^‡^	39 (21; 63)	39 (21; 62)	40 (21; 67)	0.977
Length of ICU stay (days) ^‡^	29.5 (16; 47)	29 (16; 47)	34 (17; 48)	0.396
Days on invasive mechanical ventilation ^‡^	18.5 (11; 37)	20 (11; 41)	17 (11; 32)	0.631

^†^*n* (%); ^‡^ median (first; third quartiles); ICU: intensive care unit; OLB: open lung biopsy; ARDS: acute respiratory distress syndrome; NYHA: New York Heart Association. Only Data set 1 is shown here; data sets 2–5 are reported in Appendix A.

**Table 2 jcm-08-00829-t002:** Data at the time of ARDS diagnosis and of OLB in 258 ARDS patients in the training and validation cohorts.

Day of Acute Respiratory Distress Syndrome Diagnosis					Day of Open Lung Biopsy			
	All Patients (*n* = 258)	Training (*n* = 193)	Validation (*n* = 65)	*p* Value	All Patients (*n* = 258)	Training (*n* = 193)	Validation (*n* = 65)	*p* Value
FiO_2_ ^‡^	0.8 (0.6; 1.0)	0.8 (0.6; 1.0)	0.8 (0.6; 1.0)	0.418	0.6 (0.5; 0.8)	0.6 (0.5; 0.8)	0.6 (0.5; 0.7)	0.042
PaO_2_ (mmHg) ^‡^	80 (65; 106)	78 (63; 105)	86 (68; 109)	0.342	75 (63; 89)	74 (64; 90)	76 (63; 87)	0.486
PaO_2_FiO_2_ rate ^‡^	115 (81; 160)	114 (76; 162)	124 (86; 158)	0.517	135 (93; 178)	136 (91; 178)	129 (98; 173)	0.751
PaCO_2_ (mmHg) ^‡^	41 (34; 51)	41 (35; 51)	41 (33; 52)	0.882	47 (39; 57)	47 (39; 57)	46 (39; 57)	0.814
Tidal volume (mL) ^‡^	410 (360; 455)	420 (361; 460)	400 (350; 450)	0.178	420 (359; 500)	420 (359; 500)	400 (350; 500)	0.36
Tidal volume per kg (mL/kg) ^‡^	6 (5; 8)	6 (5; 8)	6 (5; 7)	0.075	6 (5; 8)	6 (5; 8)	6 (5; 7)	0.187
Plateau pressure (cmH_2_O) ^‡^	24 (21; 28)	24 (21; 28)	24 (22; 28)	0.123	30 (26; 35)	30 (25; 33)	33 (27; 36)	0.003
PEEP (cmH_2_O) ^‡^	10 (8; 12)	10 (8; 12)	10 (8; 12)	0.044	10 (6; 12)	10 (5; 13)	10 (8; 12)	0.726
Driving pressure (cmH_2_O) ^‡^	15 (12; 19)	15 (12; 19)	14 (12; 19)	0.351	19 (16; 24)	19 (16; 24)	22 (15; 25)	0.104
Static compliance (mL/cmH_2_O) ^‡^	27 (22; 32)	28 (23; 34)	27 (19; 29)	0.012	22 (15; 27)	23 (17; 28)	16 (12; 27)	0.002
Arterial pH ^‡^	7.38 (7.29; 7.44)	7.37 (7.29; 7.43)	7.38 (7.29; 7.44)	0.369	7.39 (7.33; 7.45)	7.38 (7.32; 7.43)	7.41 (7.35; 7.46)	0.038
Respiratory rate ^‡^ (breaths per minute) ^‡^	25 (20; 30)	25 (20; 30)	25 (19; 30)	0.782	25 (20; 30)	25 (20; 30)	24 (21; 30)	0.922
Heart rate (beats per minute) ^‡^	103 (90; 120)	100 (89; 120)	111 (92; 120)	0.092	99 ± 20	98 ± 20	102 ± 21	0.172
SAP (mmHg) ^‡^	114 (100; 133)	117 (100; 134)	106 (98; 128)	0.238	119 (108; 135)	118 (106; 134)	121 (109; 140)	0.294
Temperature (°C) ^‡^	37.2 (36.5; 38.2)	37.2 (36.5; 38.0)	37.6 (36.8; 38.2)	0.210	37.0 (36.4; 37.8)	37.0 (36.4; 37.8)	37.0 (36.8; 37.6)	0.993
Prone position ^†^	2 (1)	2 (1)	0 (0)	1.000	13 (5)	9 (5)	4 (6)	0.743
Hemoglobin (gr/dL) ^‡^	10.6 (8.8; 12.2)	10.5 (8.6; 12.3)	10.7 (9.5; 12.2)	0.198	9.5 (8.6; 10.5)	9.5 (8.5; 10.5)	9.7 (8.8; 10.7)	0.163
Leukocytes (cells/µL) ^‡^	11,800 (8800; 16,300)	12,000 (8900; 15,400)	11,300 (8800; 17,600)	0.982	11,500 (8500; 17,000)	11,600 (8600; 15,800)	10,100 (6900; 18,200)	0.507
Platelets (cells/µL) ^‡^	182,000 (117,000; 282,000)	173,000 (111,000; 285,000)	189,000 (150,000; 261,000)	0.228	173,000 (111,000; 259,000)	178,000 (104,000; 263,000)	169,000 (120,000; 239,000)	0.956
Creatinine (mg/dL) ^‡^	0.88 (0.66; 1.30)	0.90 (0.70; 1.31)	0.74 (0.60;1.30)	0.046	0.88 (0.59; 1.34)	0.94 (0.62; 1.49)	0.80 (0.51; 1.13)	0.048
INR ^‡^	1.20 (0.90; 1.40)	1.20 (0.90; 1.40)	1.20 (0.90; 1.40)	0.883	1.20 (1.10; 1.30)	1.20 (1.10; 1.40)	1.20 (1.10;1.30)	0.806
Total bilirubin (mg/dL) ^‡^	0.58 (0.30; 1.06)	0.59 (0.30; 1.06)	0.50 (0.30; 0.77)	0.429	0.47 (0.24,0.90)	0.47 (0.24; 0.90)	0.40 (0.20; 0.90)	0.581
Norepinephrine (mcg/kg/min) ^‡^	0.0 (0.0; 0.1)	0.0 (0.0; 0.1)	0.0 (0.0; 0.1)	0.731	0.0 (0.0;0.1)	0.0 (0.0; 0.0)	0.0 (0.0; 0.08)	0.584
Antibiotics ^†^	180 (70)	132 (68)	48 (74)	0.502	89 (34)	71 (37)	18 (28)	0.237
Antifungal ^†^	30 (12)	24 (12)	6 (9)	0.636	21 (8)	16 (8)	5 (8)	1.000
Antiviral ^†^	29 (11)	23 (12)	6 (9)	0.714	14 (5)	10 (5)	4 (6)	0.756
Inhaled steroids ^†^	11 (4)	11 (6)	0 (0)	0.071	5 (2)	5 (3)	0 (0)	0.335
Intravenous steroids ^†^	31 (12)	23 (12)	8 (12)	1.000	116 (45)	82 (42)	34 (52)	0.218

^†^*n* (%); ^‡^ median (first–third quartiles). Only data set 1 is shown here; data sets 2–5 are reported in Appendix A. FiO_2_: Fraction of inspired oxygen; PaO_2_: arterial partial pressure oxygen; PaCO_2_: arterial partial pressure of carbon dioxide; PEEP: positive end expiratory pressure; SAP: systolic arterial pressure; INR: international normalized ratio.

**Table 3 jcm-08-00829-t003:** Bivariate analysis of comorbidities and baseline characteristics for DAD and hospital mortality.

Comorbidities	All Patients (*n* = 193)	Non DAD (*n* = 84)	DAD (*n* = 109)	*p* Value between DAD and Non-DAD	Survivors (*n*= 76)	Non Survivors (*n* = 117)	*p* Value between Survivors and Non Survivors
Active smoking ^†^	37 (19)	17 (20)	20 (18)	0.884	18 (24)	19 (16)	0.273
Active solid neoplasms ^†^	37 (19)	12 (14)	25 (23)	0.184	13 (17)	24 (21)	0.689
Active hemato-oncology neoplasm ^†^	32 (17)	15 (18)	17 (16)	0.823	8 (11)	24 (21)	0.104
Chemotherapy within the last 3 months ^†^	22 (11)	9 (11)	13 (12)	0.973	9 (12)	13 (11)	1.000
Organ transplant ^†^	7 (4)	3 (4)	4 (4)	1.000	2 (3)	5 (4)	0.706
Acute immunodeficiency syndrome/human immunodeficiency virus ^†^	7 (4)	3 (4)	4 (4)	1.000	4 (5)	3 (3)	0.437
Arterial hypertension ^†^	55 (28)	25 (30)	30 (28)	0.857	20 (26)	35 (30)	0.705
Coronary ischemia ^†^	22 (11)	3 (4)	19 (17)	0.006	9 (12)	13 (11)	1.000
Chronic cardiac failure (NYHA III or IV) ^†^	22 (11)	8 (10)	14 (13)	0.623	11 (14)	11 (9)	0.394
Chronic kidney injury ^†^	11 (6)	5 (6)	6 (6)	1.000	3 (4)	8 (7)	0.532
Diabetes mellitus ^†^	35 (18)	17 (2)	18 (17)	0.633	16 (21)	19 (16)	0.511
Diabetes mellitus requiring insulin ^†^	8 (4)	4 (5)	4 (4)	0.730	7 (9)	1 (1)	0.007
Domiciliary oxygen therapy ^†^	8 (4)	6 (7)	2 (2)	0.080	3 (4)	5 (4)	1.000
Inhaled β2 agonist ^†^	6 (3)	2 (2)	4 (4)	0.699	4 (5)	2 (2)	0.214
Inhaled steroids ^†^	11 (6)	6 (7)	5 (5)	0.537	5 (7)	6 (5)	0.755
Systemic steroids ^†^	26 (13)	13 (15)	13 (12)	0.615	10 (13)	16 (14)	1.000
Chronic obstructive pulmonary disease	35 (18)	12 (14)	23 (21)	0.303	16 (21)	19 (16)	0.511
**Baseline Characteristics**							
Female gender ^†^	66 (34)	30 (36)	36(33)	0.813	27 (36)	39 (33)	0.874
Age (years) ^‡^	61 (47)	61 (46; 69)	62 (47; 70)	0.590	56 (45; 66)	64 (49; 72)	0.010
Weight (kg) ^‡^	65 (57; 75)	64 (54; 75)	65 (58;75)	0.321	65 (56; 85)	65 (57; 74)	0.813
Hospital mortality	117(61)	43 (51)	74 (68)	0.027	35 (46)	74 (63)	0.227
Days from ARDS to OLB ^‡^	6 (2; 12)	7 (2; 13)	6 (3; 11)	0.324	6.5 (2; 11)	6 (3, 12)	0.540
Days from hospital admission to ARDS ^‡^	3 (0; 9)	3 (1; 8)	3 (0; 11)	0.897	2 (0; 9)	3 (1; 9)	0.700
Length of hospital stay (days) ^‡^	39 (21; 62)	39 (22; 57)	40 (21; 63)	0.652	47 (33; 82)	34 (17; 53)	0.000
Length of ICU stay (days) ^‡^	29 (16; 47)	29 (16; 44)	29 (16; 47)	0.751	33 (21; 57)	26 (14; 38)	0.001
Days on invasive mechanical ventilation ^‡^	20 (11; 41)	15 (11; 31)	23 (11; 43)	0.433	17 (11; 35)	21 (11; 41)	0.472

^†^*n* (%); ^‡^ median (first–third quartiles); ICU: intensive care unit; OLB: open lung biopsy; ARDS: acute respiratory distress syndrome; NYHA: New York Heart Association. Only dataset 1 is shown here; datasets 2–5 are reported in Appendix A.

**Table 4 jcm-08-00829-t004:** Bivariate analysis for DAD on the day of ARDS and the day of OLB.

	Day of Acute Respiratory Distress Syndrome Diagnosis				Day of Open Lung Biopsy			
	All Patients(*n* = 193)	Non DAD (*n* = 84)	DAD (*n* = 109)	*p* Value between DAD and Non DAD	All Patients Training Cohort (*n* = 193)	Non DAD (*n* = 84)	DAD (*n* = 109)	*p* Value between DAD and Non DAD
FiO_2_ ^‡^	0.8 (0.6; 1.0)	0.8 (0.6; 1.0)	0.8 (0.6;1.0)	0.809	0.6 (0.5; 0.8)	0.6 (0.4; 0.8)	0.6 (0.5; 0.8)	0.095
PaO_2_ (mmHg) ^‡^	78 (63; 105)	75 (62; 91)	84 (65; 115)	0.053	74 (64; 90)	74 (65; 86)	74 (63; 99)	0.770
PaO_2_FiO_2_ rate ^‡^	114 (76; 162)	107 (76; 150)	122 (78; 178)	0.188	134 (91; 178)	142 (98; 185)	132 (88; 170)	0.265
PaCO_2_ (mmHg) ^‡^	41 (35; 51)	42 (35; 50)	40 (35; 51)	0.803	47 (39; 57)	46 (38; 55)	49 (39; 58)	0.268
Tidal volume (mL) ^‡^	420 (359; 500)	420 (350; 500)	440 (360; 500)	0.368	420 (361; 460)	420 (360; 460)	402 (362; 450)	0.498
Tidal volume per kg (mL/kg) ^‡^	6 (5; 8)	6 (5; 8)	6 (5; 8)	0.564	6 (5; 8)	6 (5; 8)	6 (5; 7)	0.493
Plateau pressure (cmH_2_O) ^‡^	24 (21;28)	23 (20; 27)	25 (22; 29)	0.050	30 (25; 33)	28 (22; 32)	30 (27; 33)	0.007
PEEP (cmH_2_O) ^‡^	10 (8; 12)	10 (7; 12)	10 (8; 12)	0.292	10 (5; 13)	8 (5; 11)	10 (6; 14)	0.003
Driving pressure (cmH_2_O) ^‡^	15 (12; 19)	14 (12; 18)	17 (13; 19)	0.244	19 (16; 24)	18 (16; 23)	20 (16; 24)	0.469
Static compliance (mL/cmH_2_O) ^‡^	28 (23; 34)	28 (23; 34)	25 (21; 33)	0.244	23 (17, 28)	24 (17; 31)	23 (17; 27)	0.431
Arterial pH ^‡^	7.37 (7.29; 7.43)	7.38 (7.31; 7.43)	7.37 (7.28; 7.42)	0.509	7.38 (7.32; 7.43)	7.38 (7.34; 7.45)	7.38 (7.31; 7.43)	0.275
Respiratory rate ^‡^ (breaths per minute) ^‡^	25 (20; 30)	26 (21; 33)	24 (19; 30)	0.027	25 (20; 30)	25 (20; 30)	25 (19; 30)	0.505
Heart rate (beats per minute) ^‡^	100 (89; 120)	99 (86; 119)	102 (92; 121)	0.220	97.77 ± 20.05	96.76 ± 20.87	98.55 ± 19.46	0.544
SAP (mmHg) ^‡^	117 (100; 134)	116 (98; 132)	120 (101; 138)	0.520	118 (106; 134)	115 (106; 133)	121 (109; 135)	0.272
Temperature (°C) ^‡^	37.2 (36.5; 38.0)	37.5 (36.6; 38.23)	37.1 (36.5; 37.9)	0.182	37.14 ± 1.02	37.22 ± 1.02	37.08 ± 1.02	0.341
Prone position ^†^	2 (1)	0 (0)	2 (2)	0.506	9 (5)	4 (5)	5 (5)	1.000
Hemoglobin (gr/dL) ^‡^	10.5 (8.6; 12.3)	10.4 (8.6; 12.3)	10.6 (8.8; 12.3)	0.902	9.5 (8.5; 10.5)	9.4 (8.17; 10.1)	9.6 (8.6; 10.5)	0.144
Leukocytes (cells/µL) ^‡^	12,000 (8900; 15,400)	12,240 (9197; 15,300)	11,960 (8500; 15,590)	0.942	11,610 (8600; 15,800)	12,400 (8900; 16,625)	11,500 (8500; 15,400)	0.343
Platelets (cells/µL) ^‡^	173,000 (111,000; 285,000)	176,000 (96,000; 268,500)	173,000 (111,000; 294,000)	0.937	178,000 (104,000; 263,000)	172,000 (102,000; 277,250)	178,000 (105,000; 243,000)	0.560
Creatinine (mg/dL) ^‡^	0.90 (0.70; 1.31)	0.94 (0.72,1.38)	0.90 (0.70; 1.27)	0.675	0.94 (0.62; 1.49)	0.95 (0.63; 1.3)	0.9 (0.62; 1.71)	0.782
INR^‡^	1.2 (0.9; 1.4)	1.2 (1.0; 1.5)	1.1 (0.8; 1.3)	0.026	1.2 (1.1; 1.4)	1.2 (1.07; 1.3)	1.2 (1.1; 1.4)	0.466
Total bilirubin (mg/dL) ^‡^	0.59 (0.30; 1.06)	0.59 (0.29; 0.97)	0.60 (0.35; 1.10)	0.498	0.47 (0.24; 0.90)	0.47 (0.24; 0.90)	0.50 (0.24;0.80)	0.931
Norepinephrine (mcg/kg/min) ^‡^	0 (0,0.09)	0 (0,0.01)	0 (0,0.1)	0.151	0 (0,0.05)	0 (0,0)	0 (0,0.07)	0.582
Antibiotics ^†^	132 (68)	55 (65)	77 (71)	0.542	71 (37)	32 (38)	39 (36)	0.857
Antifungal drug ^†^	24 (12)	8 (1)	16 (15)	0.392	16 (8)	7 (8)	9 (8)	1.000
Antiviral drug ^†^	23 (12)	8 (1)	15 (14)	0.499	10 (5)	3 (4)	7 (6)	0.518
Inhaled steroids ^†^	0 (0)	0 (0)	0 (0)	-	5 (3)	0 (0)	5 (5)	0.070
Intravenous steroids ^†^	23 (12)	9 (11)	14 (13)	0.819	82 (42)	37 (44)	45 (41)	0.812

^†^*n* (%); ^‡^ median (first-third quartiles). Only dataset 1 is shown here; datasets 2 to 5 are reported in Appendix A. FiO_2_: Fraction of inspired oxygen; PaO_2_: arterial partial pressure oxygen; PaCO_2_: arterial partial pressure of carbon dioxide; PEEP: positive end expiratory pressure; SAP: systolic arterial pressure; INR: international normalized ratio.

**Table 5 jcm-08-00829-t005:** Pooled logistic regression analysis to predict diffuse alveolar damage and hospital mortality.

	At the Time of ARDS		At the Time of OLB	
Diffuse Alveolar Damage	Odds-Ratio (CI95%)	*p*	Odds-Ratio (CI95%)	*p*
Respiratory rate (breaths per minute )	0.956 (0.918–0.995)	0.029		-
Chronic coronary ischemic (reference is absence)	5.974 (1.668–21.339)	0.006	6.820 (1.856–25.061)	<0.001
PEEP (cmH_2_O)	-	-	1.131 (1.051–1.218)	<0.001
**Hospital Mortality**				
Diffuse alveolar damage (reference is absence)	2.296 (1.228–4.294)	<0.001	2.081 (1.053–4.114)	0.035
Diabetes mellitus requiring insulin (reference is absence)	0.08 (0.009–0.710)	<0.023	0.093 (0.009–0.956)	0.046
Respiratory rate (breath per minute)	1.045 (1.001–1.091)	0.046	-	-
PaCO_2_ (mmHg)	-	-	1.051 (1.019–1.084)	0.002
Platelets (cells/µL)	-	-	0.999(0.999–0.999)	0.001

ARDS: acute respiratory distress syndrome, OLB: open lung biopsy, PEEP: positive end-expiratory pressure.

**Table 6 jcm-08-00829-t006:** Summary of model’s performance for predicting Diffuse Alveolar Damage and hospital mortality.

	AUROC *	Accuracy *	Sensitivity *	Specificity *	Positive Likelihood Ratio *	Negative Likelihood Ratio *
**Day of ARDS**						
DAD (Tc)	0.660 (0.585; 0.735)	0.59 (0.52; 0.66)	0.70 (0.60; 0.78)	0.45 (0.34; 0.56)	1.28 (1.02; 1.61)	0.66 (0.46; 0.96)
DAD (Vc)	0.562 (0.417; 0.706)	0.55 (0.43; 0.68)	0.72 (0.54; 0.86)	0.41 (0.24; 0.60)	1.23 (0.85; 1.76)	0.69 (0.35; 1.36)
Mortality (Tc)	0.659 (0.583; 0.737)	0.63 (0.56; 0.70)	0.87 (0.80; 0.92)	0.26 (0.17; 0.38)	1.18 (1.02; 1.37)	0.49 (0.27; 0.90)
Mortality (Vc)	0.513 (0.361; 0.664)	0.59 (0.46; 0.71)	0.81 (0.66; 0.91)	0.21 (0.07; 0.42)	1.03 (0.80; 1.32)	0.91 (0.34; 2.48)
DAD (Lorente’s model applied to the Vc)	0.580 (0.440; 0.720)	0.48 (0.36; 0.61)	0.28 (0.14; 0.45)	0.70 (0.51; 0.85)	0.97 (0.44; 2.13)	1.03 (0.76; 1.41)
**Day of OLB**						
DAD (Tc)	0.696 (0.621; 0.769)	0.51 (0.38; 0.63)	0.66 (0.47; 0.81)	0.35 (0.19; 0.55)	1.01 (0.71; 1.45)	0.97 (0.51; 1.90)
DAD (Vc)	0.534 (0.391; 0.678)	0.51 (0.38; 0.63)	0.66 (0.47; 0.81)	0.35 (0.19; 0.55)	1.01 (0.71; 1.45)	0.97 (0.51; 1.90)
Mortality (Tc)	0.778 (0.710; 0.843)	0.72 (0.65; 0.78)	0.82 (0.74; 0.88)	0.57 (0.45; 0.68)	1.92 (1.46; 2.53)	0.32 (0.21; 0.49)
Mortality (Vc)	0.634 (0.481, 0.787)	0.70 (0.58; 0.81)	0.89 (0.76; 0.96)	0.38 (0.19; 0.59)	1.43 (1.03; 1.98)	0.29 (0.10; 0.79)
Mortality (Kao Kum’s model applied to the Vc)	0.579 (0.428; 0.730)	0.65 (0.52;0.76)	0.96 (0.84; 0.99)	0.12 (0.03; 0.32)	1.09 (0.93; 1.29)	0.35 (0.07; 2.33)

AUROC: area under receiver operator curve; * Average value of the 5 datasets. Tc: training cohort; Vc: validation cohort.

**Table 7 jcm-08-00829-t007:** Bivariate analysis for mortality on the day of ARDS diagnosis and the day of OLB.

	Day of Acute Respiratory Distress Syndrome diagnosis				Day of Open Lung Biopsy			
	All Patients (*n* = 193)	Survivors (*n* = 76)	Non Survivors (*n* = 117)	*p* Value between Survivors and Non Survivors	All Patients (*n* = 193)	Survivors (*n* = 76)	Non Survivors (*n* = 117)	*p* Value between Survivors and Non Survivors
FiO_2_ ^‡^	0.8 (0.6; 1.0)	0.8 (0.6; 0.9)	0.8 (0.6; 1.0)	0.131	0.6 (0.5; 0.8)	0.6 (0.4; 0.7)	0.6 (0.5; 0.8)	0.093
PaO_2_ (mmHg) ^‡^	78 (63; 105)	75 (60; 92)	84 (65; 118)	0.018	74 (64; 90)	74 (64; 86)	74 (64; 93)	0.784
PaO_2_FiO_2_ rate ^‡^	114 (76; 162)	104 (76; 141)	118 (83; 174)	0.131	134 (91; 178)	141 (98; 179)	132 (67; 175)	0.288
PaCO_2_ (mmHg) ^‡^	41.4 (35; 51)	41 (34; 51)	42 (35; 51)	0.458	47 (39; 57)	43 (35; 51)	52 (41; 61)	0.000
Tidal volume (mL) ^‡^	420 (359; 500)	420 (367; 500)	430 (350; 500)	0.830	420 (361; 460)	420 (379; 472)	400 (358; 450)	0.117
Tidal volume per kg (mL/kg) ^‡^	6 (5; 8)	6 (5; 8)	6 (5; 8)	0.994	6 (5; 8)	6 (5; 8)	6 (5; 7)	0.375
Plateau pressure (cmH_2_O) ^‡^	24 (21; 28)	23 (20; 27)	25 (22; 29)	0.010	30 (25; 33)	27 (22; 32)	31 (27; 35)	0.000
PEEP (cmH_2_O) ^‡^	10 (8; 12)	10 (8; 12)	10 (7; 10)	0.067	10 (5; 13)	10 (5; 13)	10 (5; 12)	0.739
Driving pressure (cmH_2_O) ^‡^	15 (12; 19)	13 (12; 17)	17 (13; 19)	0.000	19 (16; 24)	18 (14; 22)	20 (17; 25)	0.008
Static compliance (mL/cmH_2_O) ^‡^	28 (23; 34)	30 (23; 35)	25 (21; 30)	0.009	23 (17; 28)	24 (20; 34)	21 (17; 27)	0.003
Arterial pH ^‡^	7.37 (7.29; 7.43)	7.37 (7.29; 7.42)	7.37 (7.29; 7.43)	0.890	7.38 (7.32; 7.43)	7.4 (7.35; 7.46)	7.36 (7.30; 7.42)	0.000
Respiratory rate^‡^ (breaths per minute) ^‡^	25 (20; 30)	24 (18; 30)	25 (20; 30)	0.096	25 (20; 30)	23 (20; 30)	25 (20; 30)	0.117
Heart rate (beats per minute) ^‡^	100 (89; 120)	99 (89; 112)	103 (89; 123)	0.271	97.77 ± 20.05	96.26 ± 18.22	98.75 ± 21.17	0.386
SAP (mmHg) ^‡^	117 (100,134)	117.5 (102,130)	117 (99,138)	0.935	118 (106,134)	114 (105.75,134)	119 (109,135)	0.636
Temperature (°C) ^‡^	37.2 (36.5; 38.0)	37.5 (36.58; 38.32)	37.1 (36.5; 37.8)	0.234	37.14 ± 1.02	37.28 ± 1.12	37.05 ± 0.93	0.130
Prone position ^†^	2 (1)	2 (3)	0 (0)	0.154	9 (5)	5 (7)	4 (3)	0.320
Hemoglobin (gr/dL) ^‡^	10.5 (8.6; 12.3)	10.4 (8.67; 12.33)	10.6 (8.6; 12.2)	0.908	9.5 (8.5; 10.5)	9.61 (8.67; 10.5)	9.2 (8.5; 10.3)	0.174
Leukocytes (cells/µL) ^‡^	12,000 (8900; 15,400)	11,400 (8075; 15,232.5)	12,500 (9200; 17,500)	0.291	11,610 (8600; 15,800)	11,400 (8795; 15,445)	12,300 (8500; 16,000)	0.820
Platelets (cells/µL) ^‡^	173,000 (111,000; 285,000)	199,000 (117,250; 281,250)	164,000 (93,000; 285,000)	0.278	178,000 (104,000; 263,000)	213,000 (152,500; 321,250)	160,000 (80,000; 222,000)	0.000
Creatinine (mg/dL) ^‡^	0.90 (0.70; 1.31)	0.93 (0.72; 1.14)	0.89 (0.70; 1.42)	0.684	0.94 (0.62; 1.49)	0.80 (0.60; 1.18)	1.00 (0.65; 1.82)	0.014
INR ^‡^	1.2 (0.9; 1.4)	1.2 (0.7; 1.4)	1.2 (1.0; 1.4)	0.743	1.2 (1.1; 1.4)	1.2 (1.1; 1.4)	1.2 (1.1; 1.3)	0.962
Total bilirubin (mg/dL) ^‡^	0.59 (0.30; 1.06)	0.70 (0.35; 1.14)	0.53 (0.29; 0.94)	0.109	0.47 (0.24; 0.90)	0.50 (0.24; 1.00)	0.47 (0.24; 0.80)	0.142
Norepinephrine (mcg/kg/min) ^‡^	0.0 (0.0; 0.1)	0.0 (0.0; 0.1)	0.0 (0.0; 0.1)	0.739	0.0 (0.0; 0.0)	0.0 (0.0,0.0)	0 (0.0; 0.0)	0.914
Antibiotics ^†^	132 (68)	55 (72)	77 (66)	0.424	71 (37)	30 (39)	41 (35)	0.638
Antifungal drug ^†^	24 (12)	9 (12)	15 (13)	1.000	16 (8)	6 (8)	10 (9)	1.000
Antiviral drug ^†^	23 (12)	8 (11)	15 (13)	0.8	10 (5)	1 (1)	9 (8)	0.092
Inhaled steroids ^†^	-	-	-	-	5 (3)	3 (4)	2 (2)	0.384
Intravenous steroids ^†^	23 (12)	12 (16)	11 (9)	0.267	82 (42)	35 (46)	47 (4)	0.510

^†^*n* (%); ^‡^ median (first-third quartiles). Only dataset 1 is shown here; datasets 2–5 are reported in Appendix A. FiO_2_: Fraction of inspired oxygen; PaO_2_: arterial partial pressure oxygen; PaCO_2_: arterial partial pressure of carbon dioxide; PEEP: positive end expiratory pressure; SAP: systolic arterial pressure; INR: international normalized ratio.

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
