# Peer review of "Predicting the Impact of Diffuse Alveolar Damage through Open Lung Biopsy in Acute Respiratory Distress Syndrome—The PREDATOR Study"

_jcm, 2019, doi:10.3390/jcm8060829_

Reviewer 1 Report

The authors present a sound study with logically and statistically sound parameter design. The findings highlight the complexity of DAD. The paper would be strengthened with an enhanced discussion highlighting options for further study and more comparisons with literature. 

Author Response

The authors present a sound study with logically and statistically sound parameter design. The findings highlight the complexity of DAD. The paper would be strengthened with an enhanced discussion highlighting options for further study and more comparisons with literature

Thank you for this comment. In the R1, we have strengthened the discussion with a special emphasis in the relevance of ARDS and DAD as a unique clinical-pathological entity. Likewise, we compared our results with others based on autopsy, we analyzed surrogate biomarkers for DAD and why DAD could be the variable that explain the well documented endophenotypes discovered by Calfee CS et al. Finally, a new article that include patients with ARDS an OLB was quoted.

Reviewer 2 Report

Authors conducted the retrospective study by collecting data from 5 institutions including previously published data. They aimed to predict the impact of diffuse alveolar damage (DAD) in patients with ARDS by patient characteristics or respiratory and biological variables. The resource of this disease is very limited and they used multiple collaborated institutional data and managed the analysis by particular statistical methods. They concluded that clinical parameters can’t predict DAD and the DAD is associated with hospital mortality. However, this paper has several problems to be published.

Major comments

1.      Authors conclusions do not have strong evidence. Although they showed many statistical data, it is not clear which data are responsible for their conclusions.

2.      According to Table 5, does chronic coronary ischemia predict DAD? Does diabetes mellitus requiring insulin predict hospital mortality? If so, more detail discussion should be given.

3.      They made many AUROCs. What do those data mean? Are there any relations between AUROC data and DAD or mortality?

Minor comments

1.      In abstract, DMIR and AUROC should be spelled out.

2.      In methods, which institutions did agree with this study?

3.      Line 96, there are some grammatical errors.

4.      Somehow there is no data of table 2.

5.      Line 145-146, the explanation is wrong.

6.      Line 201, the space is needed between ‘the’ and ‘performance’.

7.      Line 219, there are some grammatical errors.

Author Response

Major comments

1.      Authors conclusions do not have strong evidence. Although they showed many statistical data, it is not clear which data are responsible for their conclusions.

Thank you for this comment. Our main findings are that DAD cannot be predicted clinically and that DAD is an independent factor of hospital mortality. These findings are based on the comprehensive statistical analysis we used, which may be difficult to follow even though we tried to explain it as clearly as possible. The conclusions of our study are summarized in terms of area under curves and pooled logistic regression analysis. Basically, we deciphered the following steps: 1) imputation procedure to manage the missing data; this generated 5 datasets; 2) multivariate pooled regression models; 3) area under curve to summarize the performance of the models. The difficulty was that we created five different scenarios with the aim to maintain the uncertainty of the imputation process and the multivariate analysis. For performing the multivariate analysis, we included all previously selected variables and then the program automatically applies an algorithm to all five sceneries and retrieve a unique result that include information from all sceneries. For characterizing the model (e.g. sensitivity, specificity, AUROC, etc.) the program lack of an automatic procedure, thus we had to apply the model to each dataset (from 1 to 5) and then calculate the average of each value. This is the reason why we have plotted five AUROC for each PLRM in the figure 1.

We have re-written the conclusion including into brackets the source from each affirmation: “The main findings of the present study were that DAD could not be predicted clinically (only AUROCs using the trainee cohort are better than chance) and was significantly associated with hospital mortality (DAD is associated to an increased risk of death in PLRM from the ARDS and OLB day). “

2.      According to Table 5, does chronic coronary ischemia predict DAD? Does diabetes mellitus requiring insulin predict hospital mortality? If so, more detail discussion should be given.

We agree and thank you suggestion. In the R1 we highlighted that there are well known risk and protective factors for ARDS but this is the first time that factors associated to DAD are described. We also accept that this is an epidemiology study thus is difficult to withdraw causative conclusions.

3.      They made many AUROCs. What do those data mean? Are there any relations between AUROC data and DAD or mortality?

Thank you for this comment. This issue has also been raised by another reviewer and we referred to the answer we did: “Basically, we deciphered the following steps: 1) imputation procedure to manage the missing data; this generated 5 datasets; 2) multivariate pooled regression models; 3) area under curve to summarize the performance of the models. The difficulty was that we created five different scenarios with the aim to maintain the uncertainty of the imputation process and the multivariate analysis. For performing the multivariate analysis, we included all previously selected variables and then the program automatically applies an algorithm to all five sceneries and retrieve a unique result that include information from all sceneries. For characterizing the model (e.g. sensitivity, specificity, AUROC, etc.) the program lack of an automatic procedure, thus we had to apply the model to each dataset (from 1 to 5) and then calculate the average of each value. This is the reason why we have plotted five AUROC for each PLRM in the figure 1.”.Minor comments

1.      In abstract, DMIR and AUROC should be spelled out.

Thank you. Correction made in the R1.

2.      In methods, which institutions did agree with this study?

In the R1, the name of the institution whose ethic committee approved the study was included.

3.      Line 96, there are some grammatical errors.

Thank you. Correction made in the R1.

4.     Somehow there is no data of table 2.

Table 2 contains the data at the time of ARDS diagnosis and of OLB in the training and validation cohorts. As we have to submit the paper in a compressed format and the pdf cannot be checked, unfortunately table 3 and table 2have overlapped in the pdf you have reviewed. Please, kindly accept our apologizes for the mistake. In the R1, all tables and figures are separated by a blank page.

5.      Line 145-146, the explanation is wrong.

Thank you for this comment. In the R1, we have rewritten the sentence.

6.      Line 201, the space is needed between ‘the’ and ‘performance’.

Thank you. Correction made in the R1.

7.      Line 219, there are some grammatical errors.

Thank you. In the R1 we have made every effort to correct grammatical errors.

Round  2

Reviewer 2 Report

I understand that the research for this kind of disease is very difficult and troublesome. Authors made a great effort to resolve such problems. They could not resolve the first end point, the clinical factors to detect DAD. However, the revision paper described clearly that conclusion based on their data. So readers will recognize the results and limitation.